# Investigation on the Formation Mechanism of Crack Indications and the Influences of Related Parameters in Magnetic Particle Inspection

**Long Li [1], Yun Yang [2],\*, Xiang Cai [1] and Yihua Kang [1]** 

[1]  State Key Lab of Digital Manufacturing Equipment and Technology, Huazhong University of Science and Technology, Wuhan 430074, China; louiselee@hust.edu.cn (L.L.); cai_xiang@hust.edu.cn (X.C.); yihuakang@hust.edu.cn (Y.K.)

[2]  College of Mechanical Engineering, Donghua University, Shanghai 201620, China

\*  Correspondence: yun@dhu.edu.cn

**Featured Application: The formation mechanism of crack indications and the influences of related parameters in magnetic particle inspection were investigated by theoretical analysis and an experiment, both of which provide a valuable reference for the detecting reliability improvement and the quantitative detection of cracks in automatic magnetic particle inspection.**

**Abstract:** The recent rapid development of industrial cameras and machine learning has brought new vitality to the very traditional flaw detection method, namely, magnetic particle inspection (MPI). To fully develop automatic fluorescent MPI technology, two main issues need to be solved urgently—the lack of theoretical analysis on the formation of the crack indications, and quantitative characterization methods to determine the crack indications. Here, we carry out a theoretical analysis and an experimental approach to address these issues. Theoretical models of the acting force of the leakage magnetic field were established. Subsequently, the impacts of different magnetic field strengths (1000–9000 A/m) and magnetic particle concentrations (0.5–30 mL/L) on the adsorption critical distance were analyzed. The models were solved by numerical calculations in MATLAB. In addition, a single variable control experiment was conducted to study the effects of crack images. In order to determine the quality of the crack image, three characteristic parameters were investigated, such as indication gray scale, background gray scale, and contrast ratio, were provided. The theoretical magnetic particle concentration range provided a guidance value for automated fluorescent MPI. Experimental results revealed that the optimal magnetic particle concentration was 3–4 mL/L, and, under this condition, the contrast between the crack indications and the background of crack images was obvious.

**Keywords:** quality control; crack; detection; non-destructive testing; magnetic particle inspection; model; analysis

## 1. Introduction

Fluorescent magnetic particle inspection (MPI) is one of the most widely used nondestructive testing (NDT) technologies for the detection of surface cracks on ferromagnetic material workpieces, due to its simple operation and high sensitivity [1,2]. In traditional methods, cracks are inspected manually; hence, the visual limitation of inspectors is the bottleneck for detection efficiency, reliability, and consistency. With the unprecedented applications of industrial cameras and machine learning, the automated fluorescent MPI technology based on machine vision has attracted tremendous attention in comparison to manual fluorescent MPI based on human vision, due to its high efficiency [3,4]. For example, Pautz et al. have improved the crack clarity and recognition rate by optimizing the

hardware of fluorescent MPI equipment. However, this process still does not involve automatic detection technology, such as computer image processing algorithms [5], charge-coupled device (CCD) cameras, or optical scanning, or optical scanning together with light reception by a photomultiplier tube to collect image of the cracks, computer technology to perform image processing, and pattern recognition for crack identification [6]. In addition, Wang et al. proposed an algorithm that uses the fractal dimension of crack recognition to automatically identify specific types of crack defects [7]. Although automatic discrimination systems and related algorithms based on image processing have been continuously proposed, there are still few mature industrial applications [8–11].

Currently, the development of fluorescent MPI technology is focusing on an automatic inspection system that can record crack images and analyze the images accordingly. However, there are few works regarding the underlying working principle of the automatic inspection system and no theoretical studies on this topic were found. During a cycle of fluorescent MPI, the workpiece is sprayed with magnetic particle suspension and simultaneously energized by a magnetic field. The magnetic field at cracks leaks outside the workpiece surface, thus magnetic particles in the suspension accumulate on the cracks (this phenomenon is displayed as a crack indication under 365 nm UV light). The magnetic particle suspension for fluorescent MPI is a mixture of an aqueous carrier liquid and fluorescent magnetic particles which are evenly suspended in the carrier. The concentration of magnetic particles in the suspension has a direct effect on the results of fluorescent MPI [12]. When the magnetic particle concentration in the suspension and the magnetic field imposed on the workpieces are consistent, the developed crack indications are also consistent [13]. However, in the process of fluorescent MPI, the variation of the magnetic particle concentration is very complicated. The workpiece continuously consumes magnetic particles in the suspension; thus, the magnetic particle concentration in the suspension decreases with the increase of the number of inspected parts. The changing concentration of fluorescent magnetic particles present in the suspension causes crack indications formed by magnetic particles to change continuously, thus it directly affects the reliability, stability, and consistency of the obtained results [14,15].

Existing crack indications formed by magnetic particles have no quantitative evaluation standards, which is not conducive to the development of automated fluorescent MPI technologies [16,17]. Furthermore, there is a lack of quantitative studies that investigate the effect of magnetic particle concentration in the suspension on magnetic particle crack imaging. Therefore, in the present study, in order to quantitatively determine the quality of the crack images, a comparison of different parameters of crack indication images was presented and a set of effective quantitative evaluation parameters of crack images, such as indication gray scale value, background gray scale value, and indication/background contrast ratio, was proposed. In addition, the influences of different magnetic particle concentrations on the development of crack indications were comprehended. We built theoretical models of the acting force of the leakage magnetic field, investigated the impacts of different magnetic field strengths and magnetic particle concentrations on the adsorption critical distance, and solved models by numerical calculations in MATLAB. Based on the calculation, by using a single variable control experiment, we investigated the crack images and analyzed three characteristic parameters. Thus, we hope to provide a guide for the further development of automated fluorescent MPI.

## 2. Models and Analyses

The building and solution of the model were carried out according to the following procedure:

Step 1: Define physical parameters, including particle diameter, page height, magnetic field strength, particle density, liquid density, defect width, defect height, vacuum permeability, particle magnetic susceptibility, friction coefficient, and particle concentration;

Step 2: Define the solution parameters: Spatial range = ±8 mm, Spatial step size dx = 1 μm;

Step 3: Solution roadway: traverse all space nodes and find the node closest to the force of Equation (24), and the position of this point is used as the limit distance.

### 2.1. Static Models

In order to analyze the force on the magnetic particles present in the suspension, the magnetic field distribution on the workpiece surface, especially the magnetic field leakage distribution above a crack, was first determined. This magnetic field leakage directly determines the final aggregation of the magnetic particles present in a suspension [18]. The magnetic dipole model was used to calculate the crack magnetic field leakage distribution. Figure 1 displays the magnetic field distribution at the position $P(x, y)$ above a crack, $d_c$ is the depth and $2 w_0$ is the width of the crack. In order to make the dynamic analysis of magnetic particles simple and clear, the theoretical analysis analyzes the force on a single magnetic particle [19].

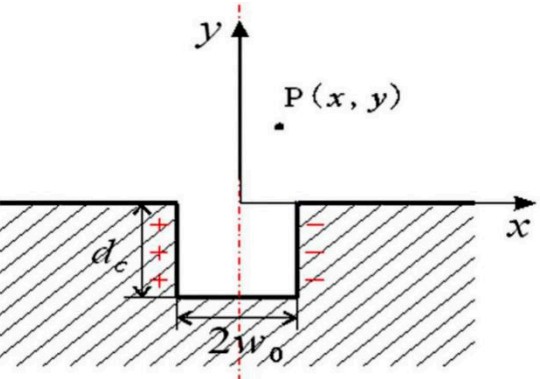

**Figure 1.** Magnetic dipole model.

First, for the case of a single magnetic particle, the force and motion of the particle in the MPI system were analyzed. Assume that the particle is a small sphere with a density $\rho$ and a diameter $d_m$. The acceleration of gravity is $g$, the density of the liquid is $\rho_w$, and the suspension height from the workpiece surface is $h_0$. As the size of each magnetic particle was small (3–8 μm; much smaller than the overall size of the research system (cm range)), the volume effect of the particle was ignored. It was also assumed that the magnetic field strength was uniform at the center of the magnetic field of the sphere.

Due to the one-to-one correspondence between the magnetic field strength and the x-coordinate, $H(x, y)$ is abbreviated as $H(x)$ in the subsequent analysis since the thickness of the magnetic particle suspension ($y$) is thin enough to allow an effect on H. The potential energy of the small particle in the magnetic field can be expressed as [20,21]:

$$U = -\int_V \frac{\mu_0 \kappa H^2(x)}{2} dx \tag{1}$$

$$H(x) = \frac{2.65\mu_0 H_w}{4\pi^2}\left[\arctan\frac{dc(x + w_0)}{(x + w_0)^2} - \arctan\frac{dc(x - w_0)}{(x - w_0)^2}\right] \tag{2}$$

where $H_w$ is the magnetic field strength, $\mu_0$ is the vacuum permeability ($4\pi \times 10^{-7}$T·m/A), and $\kappa$ is the magnetic susceptibility. The magnetic force acting on the small particles $F_m$ was represented by a negative gradient of the potential energy [21].

$$F_m = -\nabla(U) \tag{3}$$

The sphere was also subject to resistance according to the settlement dynamics:

$$F_d = C\frac{\pi}{4}d_m^2 \cdot \frac{\rho u^2}{2} \tag{4}$$

where $F_d$ is the drag force, $C$ is the drag coefficient, and $u$ is the speed of the particle.

The volume of the spherical particles is $V = \frac{\pi}{6}d_m^3$. Thus, in addition to the above forces, the magnetic particles were also subjected to gravity $F_g$ and buoyancy $F_b$ in the magnetic particle suspension carrier. They can be obtained by multiplying the product of the volume and the acceleration of gravity $g$, respectively, with the density of the magnetic particle and the liquid suspension carrier, as:

$$F_g = \frac{\pi}{6}d_m^3 \rho g \tag{5}$$

$$F_b = \frac{\pi}{6}d_m^3 \rho_w g \tag{6}$$

The fluid viscosity was also considered. Particles in the flow field were subjected to increased resistance due to Bassett forces $F_{Ba}$ [22].

$$F_{Ba} = \frac{3}{2}d_m^2(\pi\rho_w\mu)^{0.5} \cdot \int_0^t (t-t')^{-0.5}\frac{d(u)}{dt} \cdot dt' \tag{7}$$

where $t$ is the current time with the unit of second (s) and $t'$ is an integral viable with the unit of second (s).

After the particles settled on the workpiece surface, they were subjected to static friction $f_1$. The friction force experienced by an object is equal to the positive pressure multiplied by the friction coefficient, and the positive pressure equals to gravity minus buoyancy, as:

$$f_1 = \frac{\pi}{6}d_m^3 g(\rho - \rho_w) \cdot K_1 \tag{8}$$

where $K_1$ is the coefficient of friction between magnetic particle and workpiece surface.

The movement of magnetic particles in the suspension during an MPI cycle occurred in two stages: the first stage was the sedimentation of the particles suspended in the liquid onto the workpiece surface (Figure 2a), the effects of the magnetic field were omitted in order to simplify the calculation, and the second stage was the sliding of the particles from the workpiece surface toward the crack opening (Figure 2b), which involved the attraction by the magnetic field and the force of friction acting on the particles.

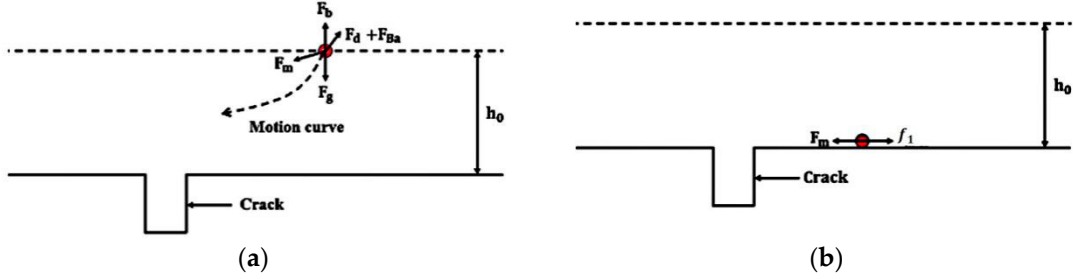

**Figure 2.** Schematic diagram of the magnetic force in (**a**) the sedimentation stage and (**b**) the sliding stage.

As the above processes involve variable acceleration motions with multiple forces, it is impossible to obtain an analytical solution. Hence, it was assumed that the fluid resistance and the Bassett force on the magnetic particles during the motion could be ignored, because these two forces are related to speed and the physical model considered in this study is based on quasi-steady state assumptions with a speed of 0 m/s. The magnetic field intensity during the entire motion process was $H(x_2)$, where $x_2$ is the position of the x-coordinate when the static friction force of the magnetic powder particles and the magnetic attraction force are balanced. As the drag is 0 and the traction is greater than the actual value of the limit adsorption critical distance, the calculated critical distance, which is defined as the limit distance at which the magnetic particles can be adsorbed by the leakage magnetic field (particles

beyond this distance will not be adsorbed), was slightly larger than the true one. Considering the effect of gravity and buoyancy, in the case of a single particle, the critical distance $L_{lim}$ can be calculated, and the derivation process is as followed.

Because the liquid level is low, the angle between the particle position and the horizontal direction is small, and the component of the magnetic field in the $y$ direction will be much smaller than the component in the horizontal direction. For the convenience of discussion, we only consider the traction force of the magnetic field on the horizontal direction of the particles. When the particles are thrown into the liquid on the surface of the workpiece, their movement is divided into two stages: the first stage is to settle from the liquid surface to the surface of the workpiece, and the second stage is to slide on the surface of the workpiece.

For the first stage, the total vertical force received by the particles is:

$$F_y = \frac{\pi}{6}d_m^3 \rho g - C\frac{\pi}{4}d_m^2 \frac{\rho u_y}{2} - \frac{\pi}{6}d_m^3 \rho_w g + F_{Bay} \tag{9}$$

The force of the particle in the $x$ direction is:

$$F_x = F_m - F_d x - F_{Bax} \tag{10}$$

Then the particle velocity in the $x$ direction $u_x$ can be obtained according to Newton's second law:

$$u_x(t) = \int_0^t a_x dt = \int_0^t \frac{F_x}{\frac{\pi}{6}d_m^3 \rho} dt = \frac{6}{\pi d_m^3 \rho}\int_0^t F_x dt \tag{11}$$

Similarly, the speed in the $y$ direction $u_y$ can be obtained as:

$$u_y(t) = \int_0^t a_y dt = \int_0^t \frac{F_y}{\frac{\pi}{6}d_m^3 \rho} dt = \frac{6}{\pi d_m^3 \rho}\int_0^t F_y dt \tag{12}$$

According to the definition of displacement:

$$S_x = \int_0^t u_x dt \tag{13}$$

$$S_y = \int_0^t u_y dt \tag{14}$$

Assuming that the height of the surface of the workpiece is $h_0$, when the particles reach the surface of the workpiece after time $t_0$, the displacement of $s_{x1}$ occurs, and the velocity in the vertical direction becomes 0 through elastic collision, and the velocity in the horizontal direction is $u_{x0}$. Then, to discuss the limit traction distance of the particles, the friction force of the particle must be greater than the magnetic traction force, the particle starts to make variable deceleration motions, until the horizontal direction $u_x$ decreases to 0, the displacement is $s_{x2}$. Marking the horizontal coordinate as $x_2$, and the particle is in a critical state. If the maximum static friction force is greater than the current magnetic traction force, the particles will remain stationary at a speed of 0, otherwise the particles will start to accelerate again until they reach the defect.

The limit state of the second stage, can be obtained according to the force balance equation of Newton's third law when the particle is at rest:

$$f_1 = F_m = \mu_0 \kappa V H(x_2) \nabla H(x_2) \tag{15}$$

$$\frac{\pi}{6}d^3 g(\rho - \rho_w) \cdot K_1 = \mu_0 \kappa V H(x_2) \nabla H(x_2) \tag{16}$$

By reducing the volume term on both sides of the equation, we can get:

$$g(\rho - \rho_w) \cdot K_1 = \mu_0 \kappa H(x_2) \nabla H(x_2) \tag{17}$$

Under the condition of fixed defect size and magnetic charge, $H(x_2)\nabla H(x_2)$ can be considered as a function on $x$, and can be simply written as $H'(x)$, and we get:

$$H'(x_2) = g(\rho - \rho_w) \cdot K_1 / (\kappa \mu_0) \tag{18}$$

That is, as long as the static friction coefficient $K_1$ is determined, the value of $x_2$ can be obtained. The adsorption limit distance of small particles is $L_{lim} = x_2 + s_{x1} + s_{x2}$.

For the first stage: based on Newtonian mechanics on free falling objects, the settling time for the magnetic particle in the first stage $t_1$ is:

$$t_1 = \sqrt{\frac{2h_0}{g(1 - \rho_w/\rho)}} \tag{19}$$

According to the definition, the horizontal displacement can be obtained:

$$S_{x_1} = \frac{1}{2}a_x t_1^2 = \frac{1}{2}\frac{\mu_0 \kappa V H'(x_2)}{m}t_1^2 \tag{20}$$

Substituting the expression at time $t_1$ and eliminating the volume term, we get:

$$S_{x_1} = \frac{\mu_0 \kappa H'(x_2)}{\rho}\frac{h_0}{g(1 - \rho_w/\rho)} \tag{21}$$

For the second stage of uniformly decelerated motion, the displacement that occurs is:

$$S_{x2} = \frac{u_1^2}{2a_2} = \frac{\left(\frac{\mu_0 \kappa H'(x_2)}{\rho} \cdot \sqrt{\frac{2h_0}{g(1-\rho_w/\rho)}}\right)^2}{2\left(\frac{g(\rho-\rho_w)\cdot K_1 - \mu_0 \kappa H'(x2)}{\rho}\right)} \tag{22}$$

It can be further simplified as:

$$S_{x2} = \frac{(\mu_0 \kappa H'(x_2))^2 \cdot h0}{g(\rho - \rho_w)((\rho - \rho_w) \cdot K_1 - \mu_0 H'(x2))} \tag{23}$$

Then the final expression of the limit distance can be obtained as:

$$L_{lim} = x_2 + \frac{\mu_0 \kappa H'(x2)}{\rho}\frac{h_0}{g(1 - \rho_w/\rho)} + \frac{[\mu_0 \kappa H'(x2)]^2 \cdot h_0}{g(\rho - \rho_w)(g(\rho - \rho_w) \cdot K_1 - \mu_0 H'(x2))} \tag{24}$$

where $\kappa$ is magnetic susceptibility and $H'$ is the magnetic field strength at the critical distance.

The above theoretical analysis mainly focuses on the force acting on a single magnetic particle in two states: (i) the motion analysis of the magnetic particle suspended in the liquid, but not contacting the object surface under the action of the magnetic field; and (ii) the motion analysis of the magnetic particle just falling on the object under the action of the magnetic field. Assuming that magnetic particles are uniformly distributed in the suspension, the "critical distance" obtained by the model theoretically determines the optimal magnetic particle concentration in the suspension. Equation (24) was derived in MATLAB, and all constant values such as particle diameter, magnetic field intensity, particle density, liquid density, and defect size were directly substituted into the above Equation (24) to obtain the final limit distance.

In the multi-particle model, it was assumed that magnetic particles distributed in the study area had a diameter of $d$, and their density and magnetic susceptibility were the same; however, their spatial positions were different. It was also assumed that $2N$ particles were uniformly distributed on the object surface and the particle spacing was $L$. As the study area was bilaterally symmetric, the particles in left side are numbered $1'$, $2'$, $3'$, ... $N'$, and the ones in the right side are numbered 1, 2, 3, ... $N$. The particle m on the right side is the adsorption limit distance, which is shown in Figure 3. The magnetic moment of the particle $i$ can be expressed as:

$$M_i = \kappa_i H(iL), M_i' = \kappa_1 H(iL) \tag{25}$$

where $M_i$ is the magnetic moment of the $i$-th particle on the right and $M_i'$ is the magnetic moment of the $i$-th particle on the left.

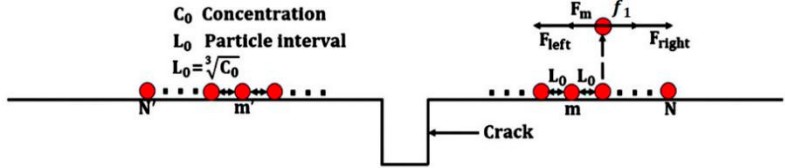

**Figure 3.** The schematic diagram of the multi-particle model.

The study of the multi-particle system was performed based on the effect of the remaining $2N$–2 particles on the particles at the adsorption limit distance, thus causing a change in $x_2$. Suppose $x_2 = mL$, it indicates that the $m$-th particle on the right side of the origin is at the critical distance. Hence, there are $n + m - 1$ particles on the left side of m and n-m particles on the right side of $m$. The magnetic field strength generated at m $H_{im}$ can be expressed as:

$$H_{im} = \frac{M_i}{4\pi r_{im}^3} r \tag{26}$$

Finally, the sum of the forces at $x_2$ can be expressed as:

$$\sum_{n'}^{n} F_m = \sum_{n'}^{n} \int_V \frac{\mu_0 \kappa_{im} H_{im}^2(r_{im})}{2} dx - \sum_{m+1}^{n} \int_V \frac{\mu_0 \kappa_{im} H_{im}^2(r_{im})}{2} dx$$
$$F_m = f^1 \max - \sum_{n'}^{n} F_m \tag{27}$$

The static friction force on the particles is fixed, and the magnetic field force on the particles is only related to the position of the particles in the static state. Therefore, when the friction force and the magnetic field force on the particles reach equilibrium, the adsorption limit distance of the particles can be derived.

*2.2. Analysis of Model Calculation Results*

A highly sensitive type of magnetic particle product, 14 A fluorescent magnetic particles, was prepared to locate very fine discontinuities in critical parts and applications. The main material of the magnetic powder was ferric oxide, as per the information provided by the manufacturer. The diameter of the magnetic particles was 8 μm, the particle density was 4800 kg/m$^3$, the volume magnetic susceptibility of the particles was $8.64 \times 10^3$, and the density of the magnetic suspension carrier was 1000 kg/m$^3$. BH curve is a curve that represents the relationship between the magnetic induction intensity B and the magnetic field intensity H in the magnetization process of ferromagnetic materials. The BH curve of the workpiece is displayed in Figure 4, where 9CrWMn steel was used, with the 4th grade of grain size, hardness of 90 HRB, surface roughness of 3.2 m, and size of 500 mm (L) × 150 mm (W) × 70 mm (D). The internal magnetic field strength and saturation magnetic

induction strength of the workpiece were 2800 A/m and 1.3 T, respectively, the crack width and height were 30 μm and 60 μm, respectively, the static friction coefficient between workpiece surface and the particles was $2 \times 10^{-7}$. The number density of magnetic particles in the suspension was converted by the magnetic particle concentration (Table 1). A pear-shape concentration measurement tube is used for measurement, the error is ±5%.

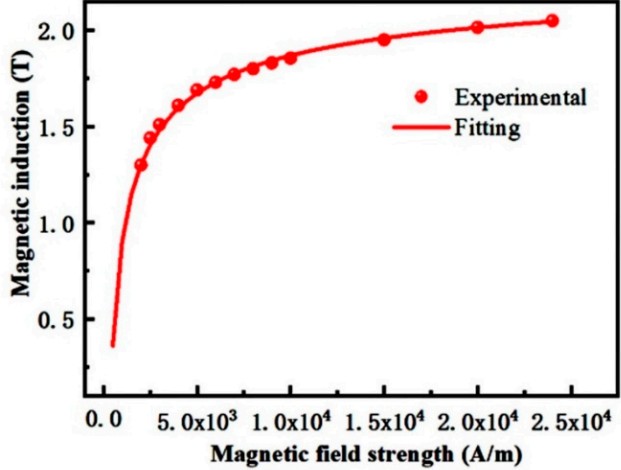

**Figure 4.** BH curve of the workpiece.

**Table 1.** Concentration and number density of magnetic particles.

| Magnetic Particle Concentration (mL/L) | Magnetic Particle Number Density ($10^7$/L) | Interval Between Particles (μm) |
|---|---|---|
| 0.50 | 18.5 | 175 |
| 1.00 | 37 | 139 |
| 2.00 | 74 | 111 |
| 3.00 | 111 | 97 |
| 4.00 | 148 | 88 |
| 6.00 | 222 | 77 |
| 10.00 | 370 | 65 |
| 20.00 | 740 | 51 |
| 30.00 | 1110 | 45 |

The influences of the magnetic field strength on the critical distance for four different magnetic particle densities (0.1 1/L, 1 1/L, 10 1/L, and $100 \times 10^7$ 1/L) are depicted in Figure 5.

The critical distance first increased and then remained unchanged with the increase of the magnetic field strength. As the effect of the magnetic force is inversely proportional to the square of the distance, the effect of the magnetic field strength will gradually decrease with the increase of the distance. With the increase of the magnetic particle concentration, the influence of multiple particles on each other increased, and the critical distance became smaller. As the magnetic field strength increased, the critical distance first increased and then remained unchanged. When the magnetic particle concentration was low, single particles and multiple particles had similar critical distance, whereas when the magnetic particle concentration increased, the result of the multi-particle situation was consistent with the actual situation.

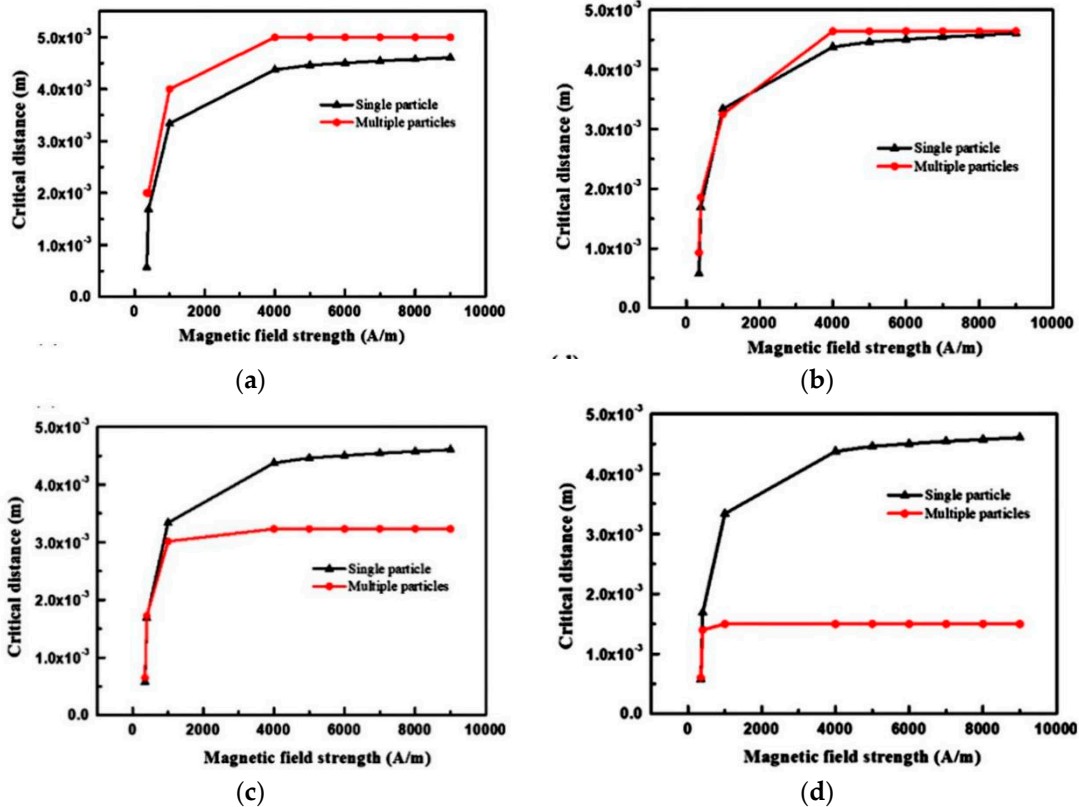

**Figure 5.** Effects of the magnetic field strength on the critical distance for four different magnetic particle densities: (**a**) $0.1 \times 10^7$ 1/L, (**b**) $1 \times 10^7$ 1/L, (**c**) $10 \times 10^7$ 1/L, and (**d**) $100 \times 10^7$ 1/L.

Figure 6 shows the effects of magnetic particle concentrations on the "critical distance" for magnetic particle accumulation. With the increase of the magnetic particle concentration, the critical distance dropped sharply and then decreased slowly. The reason for this phenomenon is that after the magnetic particles are magnetized, the magnetic field direction of the particles themselves does not completely match the leakage magnetic field formed by the simple defects on the particles around them, consequently they may be partially neutralized by each other. Therefore, as the particle concentration increases, the adsorption limit distance will decrease.

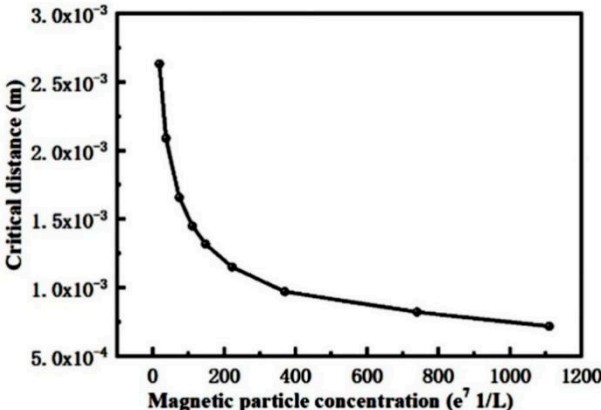

**Figure 6.** Effects of magnetic particle concentrations on the "critical distance" for magnetic particle accumulation.

Figure 7 reveals that the number of magnetic particles within the critical distance followed an approximately linear increasing trend with the increasing magnetic particle concentration, being the

number of the particles equal to the product of the adsorption limit distance and the particle concentration. Although the increase of the magnetic particle concentration caused the critical distance to shrink, its effect was less severe than that of the increasing concentration. The theoretical analysis expresses that the higher the magnetic particle concentration, the larger the number of particles within the critical distance and the brighter the magnetic particle crack images. It should be noted that the theoretical model ignores the kinetic effect and the volume effect; thus, the magnetic particles within the critical distance are stacked after being adsorbed. As a result, the effect of the actual magnetic particle concentration may not be an approximately monotonous increase, as shown in Table 1. Additionally, since with the increase of the concentration of magnetic particles, the brightness of the background of the image will increase, the overall contrast between crack indications and background may be reduced, resulting in difficulties of image recognition. Therefore, it is difficult to conclude from Figure 7 that the increase in the concentration of magnetic particles will improve the recognition ability of the magnetic particle indication image.

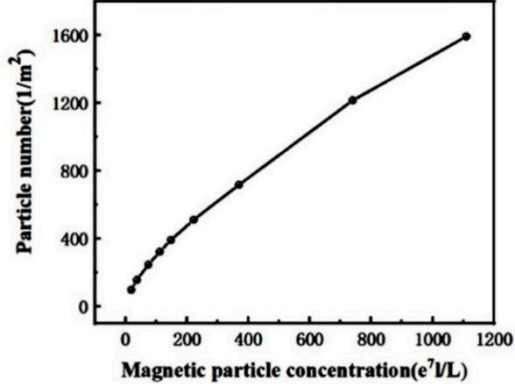

**Figure 7.** Effect of magnetic particle concentration on the number of particles.

## 3. Experiments, Results and Discussion

### 3.1. Experimental Equipment

In order to study the influences of different magnetic particle concentrations on the development of magnetic particle indication images, a test bench was designed for the evaluation of magnetic particle indications (Figure 8). The test bench consisted of a magnetization system UniMT–4000, a high-definition camera system, a magnetic particle suspension spray system, a computer and image processing system, a large-area uniform 365 nm ultraviolet lamp, a crack test piece, fluorescent magnetic particles, and related accessories.

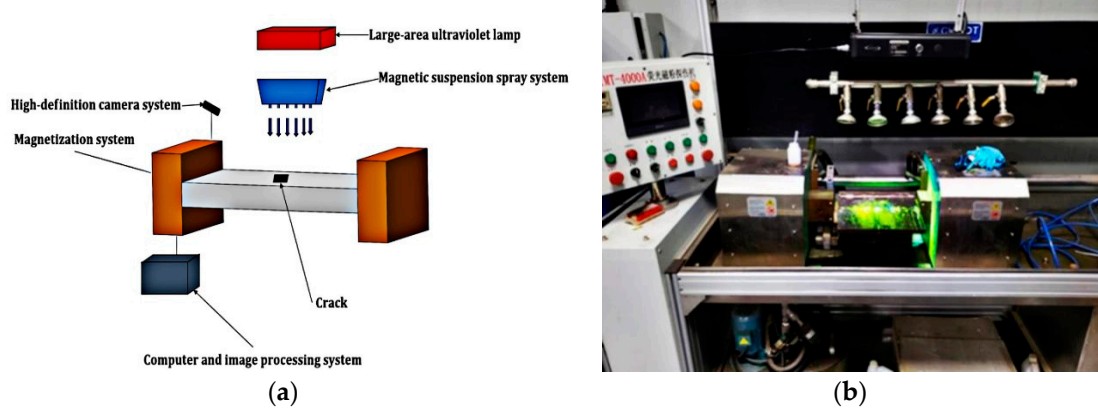

(**a**)　　　　　　　　　　　　　　　　　　　　　　　(**b**)

**Figure 8.** Schematic diagram (**a**) and photo (**b**) of the evaluation test bench for magnetic particle indications.

In the performed experiment, the control variable method was employed. Under the same magnetization conditions and considering no impurity doping in the magnetic particle suspension, a batch of magnetic particle suspensions with different concentrations was prepared to obtain crack images. The captured images of magnetic particle indications were preprocessed to extract different evaluation parameters, such as indication length, indication length/crack length ratio, gray value, and contrast [19,23].

A UniMT–4000 fluorescent magnetic particle inspection system was used to perform fluorescent MPI on the cracked test piece. This equipment used a programmable logic controller (PLC) to control the sequence of magnetic particle suspension spraying, magnetization, and image acquisition, to ensure consistent inspection conditions. The cracked test piece was made according to the high-sensitivity test piece specified in the GB/T 23907-2009 "Nondestructive Testing-Shims for magnetic particle testing" standard. The test piece material was annealed ferromagnetic pure iron with a side length of 20 mm and a thickness of 100 μm. The size of the crack was 6 mm (length) × 30 μm (depth) × 60 μm (width). According to the requirements of the ISO 9934.1–2016 "Nondestructive Testing Magnetic Particle Testing Part 1: General Principles" standard, the magnetic field strength on the workpiece surface was set to >2000 A/m. The magnetizing current of the UniMT–4000 fluorescent MPI equipment was measured by a Tesla meter placed on the surface of the cracked test piece, so that the magnetic field strength at the crack was measured to ensure that it meets the magnetization requirements of magnetic particle inspection.

### 3.2. Effects of Magnetic Particle Concentration on Magnetic Particle Indication Images

In order to systematically investigate how different magnetic particle concentrations in the suspension influence on the captured images, the magnetic particle suspension was prepared at both higher and lower concentrations than the recommended one by the MPI international standards ISO 9934, so as to determine the optimal magnetic particle concentration range [24]. Magnetic particle suspensions with the concentrations of 0.50 mL/L, 1.00 mL/L, 10.00 mL/L, and 40.00 mL/L were used as the test samples. According to the standard and practical experience, the general magnetic powder concentration is usually 3–6 mL/L. In this study, the concentration is far beyond the standard stipulated concentration. The workpiece magnetic field strength was 2400 A/m. The corresponding magnetic particle indication images captured by a high-resolution CCD are displayed in Figure 9.

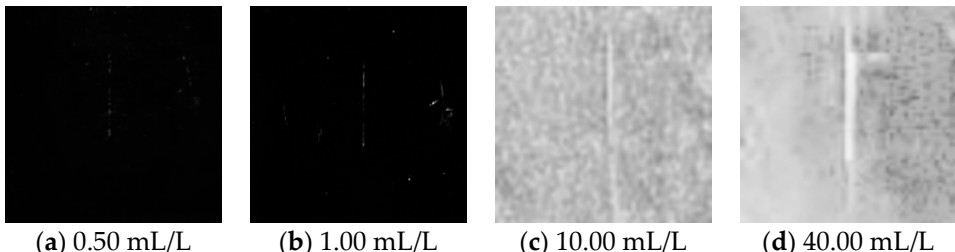

(**a**) 0.50 mL/L　　(**b**) 1.00 mL/L　　(**c**) 10.00 mL/L　　(**d**) 40.00 mL/L

**Figure 9.** Photographs of magnetic particle indications at different magnetic particle concentrations: (**a**) 0.50 mL/L, (**b**) 1.00 mL/L, (**c**) 10.00 mL/L, (**d**) 40.00 mL/L.

It is clear from Figure 9a,b that when the concentration of fluorescent magnetic particles in the suspension was very low (0.50 mL/L and 1.00 mL/L), few magnetic particles accumulated on the cracks of the workpiece. Therefore, magnetic particle indication images were blurry, the brightness of the indication was very low, the background fluorescence was weak, and the cracks were difficult to identify [25]. On the contrary, when the concentration of fluorescent magnetic particles in the suspension was very high (10.00 mL/L and 40.00 mL/L), due to the presence of a large amount of magnetic particles on the test piece surface, the background fluorescence was too strong, the magnetic particle indication on the crack had no obvious boundary with the background, and the contrast of the crack indications against the background, therefore, the cracks were difficult to identify. The above

two types of extreme magnetic particle concentrations were not conducive to differentiating crack indications from the captured images [26–28].

The recommended values specified in MPI international standards are mostly empirical values based on human observation, and no quantitative method is available to determine the appropriate magnetic particle concentration in the suspensions [3,29,30]. Therefore, in order to revise the magnetic particle inspection standard, reveal the influences of magnetic particle concentrations in the suspension on the quality of crack indications, and guide the development of automated MPI technologies, it is necessary to comprehensively and systematically analyze the changes of the magnetic particle concentration in a suspension during the inspection process. In the present experiment, a controlled variable method was used for this purpose.

The gray scale values of the crack images were analyzed in MATLAB so as to obtain different values of magnetic particle indication image evaluation parameters, such as indication gray scale value as well as background gray scale value, and the corresponding results are summarized in Table 2. The definition of the image contrast $C_j$ is as:

$$C_j = \sqrt{\frac{\sum_{i=1}^{N}(x_i - \bar{x})^2}{N}} \tag{28}$$

where $N$ is the number of pixels, $x_i$ is the grayscale of the sampling point, and the $\bar{x}$ represents the average gray scale value.

When the magnetic particle concentration in the suspension was low (1–15 mL/L), it is clear from Figure 10 that as the magnetic particles got higher, all parameters, including the gray scale value of crack indication and the background gray scale value increased rapidly. When the magnetic particle concentration in the suspensions was in the range of 15–50 mL/L, the background gray scale value increased gradually and reached the threshold of 248, the maximum value in Table 2. It should be noted that the contrast ratio value increased first and then decreased in the magnetic particle concentration range of 1–15 mL/L. The contrast ratio reached the peak when the concentration of the magnetic particle suspension was 3 mL/L and then started to decrease with the increase of the magnetic particle concentration. When the concentration of the magnetic particle suspension reached 10 mL/L, the contrast ratio tended to be consistent. The conditions for automatic defect recognition (ADR) are different from but similar to that for manual MPI. In this study, the main focus is on the three characteristic parameters, namely, indication gray scale, background gray scale, and contrast ratio. For an ADR process, these three parameters can be easily set and adjusted so that to produce a satisfactory image in a manual MPI, the image quality is largely dependent on the operators' experience. Therefore, by using these quantitative parameters, ADR can offer better image quality.

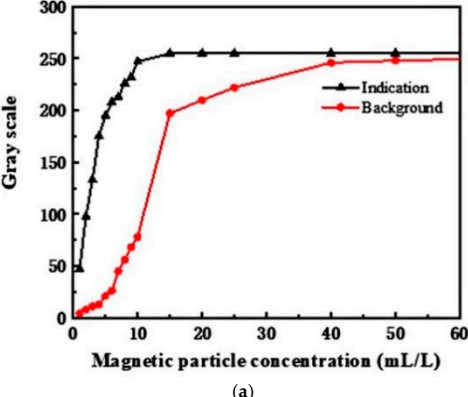

(a)

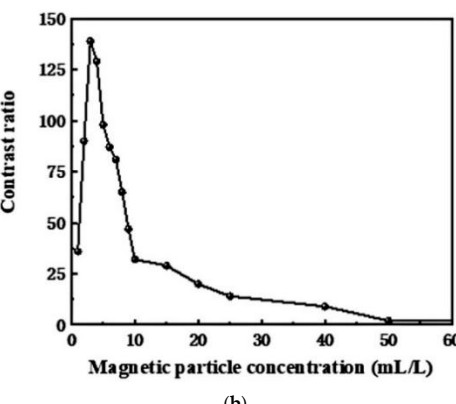

(b)

**Figure 10.** (**a**) Magnetic particle indication gray scale and background gray scale curves and (**b**) Contrast ratio curve.

**Table 2.** Evaluation parameters under different magnetic particle concentrations.

| Concentration mL/L | 1.00 | 2.00 | 3.00 | 4.00 | 5.00 | 6.00 |
|---|---|---|---|---|---|---|
| Gray scale image |  |  |  |  |  |  |
| Indication gray scale | 47 | 98 | 133 | 175 | 195 | 208 |
| Background gray scale | 4 | 8 | 11 | 13 | 21 | 26 |
| Contrast ratio | 36 | 90 | 139 | 129 | 98 | 87 |
| Concentration mL/L | 7.00 | 8.00 | 9.00 | 10.00 | 15.00 | 20.00 |
| Gray scale image |  |  |  |  |  |  |
| Indication gray scale | 213 | 226 | 232 | 247 | 255 | 255 |
| Background gray scale | 45 | 56 | 68 | 78 | 197 | 210 |
| Contrast ratio | 81 | 65 | 47 | 32 | 29 | 20 |
| Concentration mL/L | 25.00 | 40.00 | 50.00 | | | |
| Gray scale image |  |  |  | | | |
| Indication gray scale | 255 | 255 | 255 | | | |
| Background gray scale | 222 | 246 | 248 | | | |
| Contrast ratio | 14 | 9 | 2 | | | |

Therefore, the optimal magnetic particle concentration was found to be in the range of 3–4 mL/L. Moreover, when the magnetic particle concentration in the theoretical model increased, the number of particles in the critical distance increased monotonically. A critical value was noticed when the magnetic field strength of the workpiece reached 2400 A/m, therefore, the influences of kinetic effects, the change of the background magnetic field after stacking, and the change of the number of particles adsorbed in the range within the adsorption limit distance should be explored. In real conditions, a too high or too low magnetic of a particle concentration is not conducive to obtaining the best magnetic particle indication images. When the magnetic particle suspension concentration was too low (<3 mL/L), very few magnetic particles were available to form an effective magnetic particle indication image in the cracked area, on the other hand, when the magnetic particle suspension concentration was too high (>4 mL/L), due to the continuous increase of the magnetic particle density, the magnetic particle suspension was transformed from the mobile fluid state with suspended magnetic particles

to the slurry state with, consequently, too many particles of extremely poor fluidity. The magnetic field leakage at the crack no longer effectively affected the movement and accumulation of magnetic particles to form the indication; thus, the signal-to-noise ratio of the image was too low.

By controlling the optimal magnetic field strength and magnetic particle concentration, the reliability and consistency of automated magnetic particle inspection can be guaranteed. After processing the image information collected by the high-resolution CCD of the high-definition camera, the change in the concentration of the magnetic powder can also be monitored to form a closed-loop control.

This article does not consider the detection conditions under extreme conditions, such as extremely low temperatures (below −20 °C) and extremely high temperatures (above 80 °C); in these cases, the viscosity cannot be ignored. This limitation shall be overcome in future studies.

## 4. Conclusions

The present study investigated the effects of different magnetic particle concentrations on magnetic particle indication images. First, a 1 D magnetic particle critical adsorption distance model was established, and the effects of the background magnetic field strength and the magnetic particle concentration on the critical adsorption distance were analyzed. Second, based on the 1 D model, the influences of the background magnetic field strength and the magnetic particle concentration on the horizontal critical adsorption distance were studied. It was verified that the critical distance increases with the increase of the magnetic field strength, and then, after overrunning the threshold of 5000 A/m, tends to be stable. Finally, an automatic flaw detection device was designed, and the effects of different magnetic particle concentrations (1–50 mL/L), under the magnetic field strength of 2400 A/m, on the gray value of magnetic images and the background gray value were discussed. From this analysis, it was found that the optimal particle concentration ranges between 3–4 mL/L. Therefore, a too high or too low of a magnetic particle concentration could significantly affect the captured magnetic particle indication images.

The proposed theoretical model can be used to study the magnetic particle concentration, which determines the distance between magnetic particles and affects the mutual attraction and repulsion between them, and the influence of the magnetic field strength on the critical distance. In the future work, the kinetic effects and the geometric effects of magnetic particles on adsorption limit distance will be discussed. The effects of the dynamic adsorption and stacking of magnetic particles on the background magnetic field will also be investigated. By controlling the optimal magnetic field intensity and magnetic particle concentration, the consistency and reliability of the crack image can be guaranteed. Moreover, after processing the image information collected by the high-resolution CCD of the high-definition camera, the change of the magnetic particle concentration can be monitored to form a closed-loop control. The diagnosis rate and efficiency are improved. Furthermore, for the process optimization, it is necessary to consider the magnetic particle suspension doping in the actual working environment and the problem of particle concentration loss; hence, the relationship between concentration optimization and compensation under actual conditions will be examined as well.

**Author Contributions:** L.L., Y.Y. and Y.K. conceived the paper structure; L.L. and Y.Y. performed the static models and analyses; L.L. and X.C. designed and conducted the experiments; L.L., Y.Y. and Y.K. prepared the manuscript and checked the writing. All authors have read and agreed to the published version of the manuscript.

**Funding:** This research was funded by the National Natural Science Foundation of China (NNSFC) (No. 51807022, No. 51875226) and the Fundamental Research Funds for the Central Universities (No. 2232018D3-26).

**Conflicts of Interest:** The authors declare no conflict of interest.

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
