# Peer review of "Investigation on the Formation Mechanism of Crack Indications and the Influences of Related Parameters in Magnetic Particle Inspection"

_applsci, doi:10.3390/app10196805_

Round 1

Reviewer 1 Report

The authors presented the results of the analysis of the influence of various factors defining measurement conditions on the subsequent later effectiveness of surface defects detection in magnetic elements by means of magnetic particle inspection method. The results presented by authors may be of interest of the readers in the field. However, the way of presentation and description in many parts of the paper is not sufficient and a bit enigmatic. The authors have been pointing out model defining relationship without referring to the background knowledge of the presented dependencies, what makes the reader hard to validate the correctness and follow the procedure as well. The publishing of the paper can be considered, however the revision of the presentation way must be incorporated first. Below there are given some general and detailed remarks as well.

General remarks

  1. Although the introduction section gives some background on the current problems with MPI, however there is lack of the general description of the procedure which the authors followed solving the problem. The authors should briefly but clearly state at the end of the introduction section what the problem is, how they want to achieve the main goal of the paper and why. The following steps of the presentation and theirs motivation should also be addressed what would allow to reach the general overview of the work before going into details and clarify the authors work to the reader.
  2. In the introduction sections authors refers to the requirements for the MPI imaging technology in the context of the automatic defect recognition (ADR) needs. However, actually in the paper there is only reference to the achieved quality of images (gray level or contrast to noise ratio assessment) but they give no comments on how the chosen conditions could affect the effectiveness of the automatic defect detection techniques. It should at least be more deeply discussed.
  3. The description of the model is not clear and should be improved. The authors do not derive the presented dependencies themselves, which is off course not required, but when authors provide them in the paper, they do not refer to the sources in any way - it is not clear from what they arise. Therefore authors should indicate the source of the dependencies used or precisely justify their formulas. There is often a lack of a clear description of all variables as well: for example, in line 98, the authors refer to the the formula for potential energy and then provide two relationships without pointing the exact one; similarly, in (4), what C and u stands for? Finally there is not much about the implementation of the model in Matlab software and the details should be supplemented.
  4. It would be beneficial to introduce at the beginning of section 2 the schematic diagram of the whole calculation procedure, depicting the successive stages of the procedure and showing the relationship between the parameters, what would allow the reader to clearly understand the model construction.

Detailed remarks:

  1. In the Abstract section the sentence in lines 22-25 should be split into two sentences. Please correct the comma in line 24 placement as well.
  2. The font size in the figures are varying two much within the whole paper. The size used in figure 2 is too small, as the symbols depicted are barely visible.
  3. Line 145-151: the variables should be written in italic style.
  4. Line 178: there should be a reference to Figure 5 not to figure 4.
  5. Figure 5 caption: should it be 10 to the power of 7 not 107?
  6. Lines 183-185: sentence not clearly stated
  7. Line 185-186: “the influence of multiple particles increased” – the influence on what?
  8. Lines 187-190: the meaning is not clear.
  9. Line 247-248: Authors have measured the current by teslameter?
  10. In the conclusions section authors wrote: “It was found that the optimal background magnetic field strength was between 1000-4000 A/m” but they did not refer to this in any part of the paper earlier, at least it is not clearly stated.

Author Response

Thanks!

Reviewer 2 Report

Notes in the attached file.

Author Response

Thanks!

Reviewer 3 Report

I don't know if all the figures were original or if some were obtained from other works, since there are no references on the captions I assume that they are all original, but if not please put the referece.

Author Response

Thanks!

Round 2

Reviewer 1 Report

The authors made numerous changes and responded accordingly to the remarks. Nevertheless, it is reasonable for them to review the content of the article before a possible publication, because there are phrases, especially in the corrected parts, that are not clearly formulated, e.g. the beginning of section 2 (line 88) or the end of line 138 and the beginning of 139.

Author Response

Dear Reviewer,

We are truly grateful to your kind comments and thoughtful suggestions on our manuscript. Based on these comments and suggestions, we have made careful modifications on the original manuscript. All changes made to the text are in red color. Attached document is our point-by-point responses to the comment questions. 

Thank you very much for your work concerning our paper.

Wish you all the best!
